# Methylation Patterns of the FKBP5 Gene in Association with Childhood Maltreatment and Depressive Disorders

**DOI:** 10.3390/ijms25031485

**Published:** 2024-01-25

**Authors:** Nora L. Großmann, Antoine Weihs, Luise Kühn, Susann Sauer, Simone Röh, Tobias Wiechmann, Monika Rex-Haffner, Henry Völzke, Uwe Völker, Elisabeth B. Binder, Alexander Teumer, Georg Homuth, Johanna Klinger-König, Hans J. Grabe

**Affiliations:** 1Department of Psychiatry and Psychotherapy, University Medicine Greifswald, 17475 Greifswald, Germany; 2German Center for Neurodegenerative Diseases (DZNE), Site Rostock/Greifswald, 17489 Greifswald, Germany; 3Department Genes and Environment, Max Planck Institute of Psychiatry, 80804 Munich, Germany; 4Institute for Community Medicine, University Medicine Greifswald, 17475 Greifswald, Germany; 5German Centre for Cardiovascular Research (DZHK), Partner Site Greifswald, University Medicine Greifswald, 17475 Greifswald, Germany; 6Interfaculty Institute of Genetics and Functional Genomics, University Medicine Greifswald, 17475 Greifswald, Germany; 7Department of Psychiatry and Behavioral Sciences, Emory University School of Medicine, Atlanta, GA 30322, USA

**Keywords:** childhood maltreatment, depression, HPA, epigenetics, stress, cortisol, genetics

## Abstract

Childhood maltreatment is an important risk factor for adult depression and has been associated with changes in the hypothalamic pituitary adrenal (HPA) axis, including cortisol secretion and methylation of the *FKBP5* gene. Furthermore, associations between depression and HPA changes have been reported. This study investigated the associations of whole-blood *FKBP5* mRNA levels, serum cortisol levels, childhood maltreatment, and depressive symptoms with the whole-blood methylation status (assessed via target bisulfite sequencing) of 105 CpGs at the *FKBP5* locus using data from the general population-based Study of Health in Pomerania (SHIP) (*N* = 203). Both direct and interaction effects with the rs1360780 single-nucleotide polymorphism were investigated. Nominally significant associations of main effects on methylation of a single CpG site were observed at intron 3, intron 7, and the 3′-end of the gene. Additionally, methylation at two clusters at the 3′-end and intron 7 were nominally associated with childhood maltreatment × rs1360780 and depressive symptoms × rs1360780, respectively. The results add to the understanding of molecular mechanisms underlying the emergence of depression and could aid the development of personalised depression therapy and drug development.

## 1. Introduction

Childhood maltreatment (CM) has been associated with mood and anxiety disorders during adulthood [1,2,3,4,5,6]. According to the vulnerability–stress model of depression, both genetic predispositions and environmental stressors contribute to the emergence of the disease [7]. Although genetic studies have found evidence for moderate heritability of depressive disorders, no single genetic factor could be linked to the development of depression [2,8].

Epigenetic mechanisms such as DNA methylation potentially manifest environmental experiences in biological changes and therefore may be involved in the pathogenesis of depression [2]. Recent studies reported the long-lasting effects of traumatic experiences on *FKBP5* methylation [9,10,11,12]. FKBP5 is a co-chaperon of the glucocorticoid receptor (GR) complex, which lowers the GR affinity to cortisol binding [13] and delays nuclear translocation [14]. While FKBP5 impedes the GR activity, the translocated GR complex serves as a strong enhancer for *FKBP5* transcription. Thus, *FKBP5* is part of an ultra-short negative feedback loop for cortisol secretion within the hypothalamic pituitary adrenal (HPA) axis, which is the major regulatory system of the endocrine stress response [13,15]. Furthermore, it is being investigated whether early trauma or depression are associated with alterations in basal cortisol levels [16,17,18].

The present manuscript focused on alterations of cortisol and *FKBP5* since early stress exposure was suggested to substantially impact psychological and biological stress response mechanisms [3,8].

From a psychiatric point of view, *FKBP5* is of particular interest. *FKBP5*-associated single-nucleotide polymorphisms (SNPs), in particular rs1360780, showed an impact on depression treatment and the recurrence of depressive episodes [16]. Moreover, the interaction between the *FKBP5* genotype and childhood abuse has been associated with the severity of present adult depressive symptoms [17], as well as structural brain changes in emotion-processing brain areas [18]. Supportively, other studies found altered *FKBP5* methylation after stressful life events in connection with risk alleles at single CpGs or predefined regions [9,10,11,12]. It was hypothesised that demethylation of *FKBP5* caused by excessive cortisol exposure during early life stress leads to long-lasting HPA axis dysregulation, enhancing the risk for stress-related psychiatric disorders in adulthood [10].

This study investigated methylation changes of the *FKBP5* gene as a possible mediator between CM and depressive symptoms in adulthood in a general population sample (Figure 1). Since altered DNA methylation may influence the transcription of *FKBP5,* which is part of the negative feedback loop for cortisol secretion, *FKBP5* mRNA levels and cortisol levels were also investigated. In particular, we examined the following hypotheses:*FKBP5* methylation levels are associated with *FKBP5* mRNA levels (Figure 1c).*FKBP5* methylation levels are associated with basal cortisol levels (Figure 1f).CM is associated with *FKBP5* methylation levels (Figure 1b).Current depressive symptoms are associated with *FKBP5* methylation levels (Figure 1g).

While previous studies mainly focused on array-based single CpG sites or pre-defined methylation regions [9,10,11,12,21], this study examined 105 methylation sites assessed via targeted bisulfite sequencing of the whole *FKBP5* locus and, in a second step, used a data-driven approach to reduce CpGs. Thus, we complemented the approach by Wiechmann et al. who reported an association of *FKBP5* methylation with depressive symptoms [22], with an overview on gene expression, cortisol levels, and childhood trauma, to explore the underlying mechanisms.

As former studies reported the moderation effects of rs1360780 [9,10,11,12], we additionally analysed the interaction between the exposure variable and the SNP for each model as sensitivity analyses.

## 2. Results

### 2.1. Sample Characteristics

In total, the current study included *N* = 203 participants from a subsample of the Study of Health in Pomerania (SHIP), referred to as SHIP-TREND, with *FKBP5* whole-gene methylation, of which *N* = 103 (50.7%) had experienced some form of CM. Psychometric scores are presented in Figure 2; CM was assessed with the Childhood Trauma Questionnaire (CTQ), and depressive symptoms were assessed with the depression module of the Patient Health Questionnaire (PHQ-9). Table 1 provides an overview of the baseline characteristics of the entire SHIP-TREND cohort and the herein analysed subsample.

### 2.2. FKBP5 Methylation

In total, 105 CpG sites of the *FKBP5* locus (chromosome 6, hg19) were measured. Since we expected non-variable sites, i.e., sites where the methylation beta values of all individuals were <20% or >80%, to be less informative, they were excluded in the present analyses, leaving 34 CpG sites to be included in this study. To provide an overview over the whole *FKBP5* gene including the 34 selected CpG sites, mean methylation levels are presented in Appendix A and correlations of methylation *M*-values are presented in Appendix A. Investigating the pairwise correlation of the methylation *M*-values across the whole *FKBP5* gene revealed mostly positive correlations among neighbouring CpG sites, while negative correlations were less frequent (Appendix A). One highly correlated cluster of 38 CpGs was located at the transcription start site (correlation median: 0.56; range: [−0.21, 0.84]), with all being non-variable sites. Other clusters formed at the 5′-end, in intron 5 and at the 3′-end.

### 2.3. Direct Effects

Results of linear mixed effect models are presented in the Appendix A. A power analysis (see Appendix A for details) revealed that, for 203 participants, only effects (Cohen’s f) above 0.1 may be detected with a power of 0.95 and significance level of 0.05/34 = 0.0014 corrected for 34 tests.

Nominally significant results are presented in Table 2 and effect sizes in Figure 3; however, none survived multiple testing correction. The largest effect sizes were detected for cortisol at the 3′ region (maximum at chr6:35704069, *β* = −0.637, *t* = −1.956, *p* = 0.052). For gene expression, the largest effect was observed in intron 3 (chr6:35592135, *β* = 0.188, *t* = 1.715, *p* = 0.088), and for CTQ, in the 3′ region (chr6:35693881, *β* = −0.257, *t* = −1.724, *p* = 0.086). In the PHQ-9 model, three neighbouring CpGs in the 5′ region tended towards a positive association (chr6:35490654, chr6:35490693, and chr6:35490744, with effect sizes between 0.057 and 0.076).

### 2.4. Interaction Effects

We additionally investigated possible interaction effects between the SNP rs1360780 and CM on gene methylation. However, after correction for multiple testing no *p*-value was below the significance threshold 0.05/34 = 0.0015 (Appendix A). CpGs with nominal *p*-values < 0.05 are listed in Table 3 and their observed effects are presented in Figure 4 and Figure 5. In particular, our data hinted at an interaction effect of CTQ and the SNP on methylation at the 3′-region of *FKBP5* and an interaction of PHQ-9 and rs1360780 on methylation in intron 7 (Table 3). No CpGs were nominally significant in the interaction of genotype and methylation on gene expression or cortisol.

## 3. Discussion

We hypothesised that *FKBP5* methylation mediates the link between CM and adult depression by altering *FKBP5* expression and thus the HPA axis stress response, and that this relationship is influenced by rs1360780. Therefore, we investigated 34 CpGs along the *FKBP5* gene in association with CM, *FKBP5* mRNA levels, cortisol, and current depressive symptoms in 203 participants of the general population-based SHIP-TREND baseline cohort. Likely due to the small sample, none of the observed effects were statistically significant after correction for multiple testing. Figure 6 provides an overview of nominally significant findings. As, to the best of our knowledge, only few studies have analysed methylation data of the entire *FKBP5* gene, all obtained results are reported enabling inclusion in future meta-analyses and to point out promising CpG sites for future research beyond the ones typically analysed.

### 3.1. FKBP5 Methylation and FKBP5 Expression

Altered *FKBP5* gene expression, regulated by DNA methylation, is hypothesised to disrupt the HPA feedback loop and enhance the risk for stress-related psychiatric disorders [10,23]. Klengel et al. observed that lower methylation in intron 7 enhanced *FKBP5* transcription however, only after GR activation but not under basal conditions [10]. They hypothesised that increased FKBP5 induction tightens the feedback loop and leads to GR resistance [10], leading to prolonged stress response and increased risk for stress-related psychiatric disorders [13]. However, using basal *FKBP5* mRNA levels, the current analysis did not detect any significant direct effects of *FKBP5* methylation on gene expression, which is in line with the results of Klinger-König et al. [11]. Furthermore, intron 7 shows no consistent effect direction.

### 3.2. FKBP5 Methylation and Cortisol

Glucocorticoid treatment reduced *FKBP5* methylation in human and rodent cells [10,22,24,25]. Presumably, glucocorticoids activate base excision repair mechanisms where methylated cytosine is replaced by unmethylated cytosine [15].

In this analysis, no differentially methylated CpG sites at the *FKBP5* locus were observed for serum basal cortisol levels.

Klengel et al. hypothesised that in risk allele carriers of the *FKBP5* SNP rs1360780, elevated cortisol levels induce changes in *FKBP5* methylation that increase the GR activation bound to FKBP5, tightening the feedback loop and leading to glucocorticoid resistance [10]. Nevertheless, they could not observe any association between *FKBP5* methylation and current cortisol levels [10]. Again, this might be limited to stress responses as demonstrated by Höhne et al. [9] and is thus not observable under basal conditions.

Aside from this, Klengel et al. suggest that a genotype-mediated increase in *FKBP5* transcription levels hampers the feedback loop and leads to a prolonged cortisol release due to GR resistance [10]. Methylation-related alterations of the feedback loop could have a similar effect. However, since we used basal cortisol levels, we could not investigate alterations in stress response.

### 3.3. FKBP5 Methylation and Psychopathology

The HPA system is developing during childhood and both basal rhythms and reactivity are shaped by experience [26]. In rodents, the number and amplitude of cortisol peaks during the circadian cycle increased after early life exposure to endotoxins [27]. In humans, early stress exposure can lead to elevated plasma cortisol and various health issues in adult life [20].

Altered HPA axis dynamics presumably mediate the development of depressive disorders after experiencing CM [3]. Stress during early development could change the *FKBP5* methylation, leading to a disrupted feedback loop and slower regulation of the HPA stress reactions [2,15], which could promote the development of depressive disorders [10].

In intron 7, Tyrka et al. observed lower methylation levels in two CpG sites in maltreated compared to non-maltreated children [28], and Klinger-König et al. found an association of depressive symptoms with reduced methylation for one CpG site [11]. The present analysis detected neither methylation changes in intron 7 related to CM nor depressive symptoms, aligning with the results of Bustamante et al. [21]. Furthermore, Wiechmann et al. reported an association with major depressive disorder (MDD) at intron 5, as well as the 3′ region and 5′ region of the gene [22], which was not detected in the present analysis.

### 3.4. Interaction Models

Associations between *FKBP5* methylation and psychopathology in interaction with the *FKBP5* genotype of SNP rs1360780 have previously focused on specific regions: Klengel et al. observed an interaction effect between genotype and CM on methylation in intron 7, but not in intron 2 and intron 5. In particular, methylation was reduced only in T-allele carriers with CM but not in non-trauma and/or non-carrier groups for three CpGs (chr6:35558488, chr6:35558513, chr6:35558566, hg19) [10]. Mihaljevic et al. replicated this gene–environment interaction for chr6:35558566 (hg19) in healthy individuals but not in patients with psychosis or their siblings [29]. However, in the same sample, T-allele-carrying patients had lower methylation levels than their siblings or healthy controls independent of trauma exposure. Similarly, Kang et al. observed reduced methylation in T-allele carriers independent of PTSD at chr6:35558488 and chr6:35558513 (hg19), while T-allele-carrying PTSD veterans had higher methylation levels compared to T-allele-carrying non-PTSD veterans [30]. For the same CpGs, Saito et al. observed reduced methylation in T-allele carriers with emotional abuse/neglect compared to T-non-carriers in patients with bipolar disorder but not in patients with MDD [31].

In our data set, none of the described CpGs reached significance in the interaction models after correction for multiple testing. However, lower methylation levels of two neighbouring CpGs in intron 7 (chr6:35558386 and chr6:35558438) were nominally associated with more depressive symptoms in T-allele carriers. Our results are in line with the results reported by Höhne et al., who observed slightly higher DNA methylation in individuals with remitted MDD compared to healthy controls in T-allele carriers [9]. Although not reaching statistical significance after adjustment for multiple testing, there was a cluster of three neighbouring CpG sites at the 3′-end (chr6:35694724, chr6:35694748, chr6:35694756) that were nominally associated with depressive symptoms, dependent on the rs1360780 genotype, and might be fruitful in future studies.

### 3.5. Strengths and Limitations

While former research focused on predefined genomic regions, this study examined CpGs across the whole *FKBP5* locus enabling us to detect new candidates for differentially methylated CpGs. In order to reduce multiple testing, we used a data-driven approach to select promising CpGs. Note that this does not mean that the excluded CpGs are uninformative, especially when considering other data sets.

With the rather small sample size, the power analysis indicated that only medium effect sizes (>0.1) could be detected in the reduced data set. We therefore strongly encourage further investigation of these CpGs, when a larger sample size is available and have therefore, additionally to the investigated 34 sites, included results of the remaining 71 non-variable probes in the Appendix A. Furthermore, high values of PHQ-9 and CTQ scores were rare in the analysed cohort.

The sample size also limited the complexity of our model; in future studies, a stratification for sex could investigate sex-specific differences in changes of methylation patterns.

It is possible that CM only leads to methylation changes when a genetic vulnerability is present [7]. Whereas previous studies reported a genotype-dependent *FKBP5* effect [9,10,11,12], our study focused primarily on genotype-independent effects. In a second analysis, we included the most commonly reported SNP in a sensitivity analysis. However, other genetic variants and environmental factors that we did not investigate due to a limited sample size may play a critical role.

Since Klengel et al. observed neither an influence of current cortisol levels nor adult trauma on *FKBP5* methylation, they suggested that the epigenetic changes only occur at a sensitive period during childhood [10]. Hence, cortisol effects may not be detectable in an adult population. Höhne et al. observed altered cortisol levels and *FKBP5* mRNA levels for depressed risk allele carriers after stress induction [9]. It is still being investigated whether depression or early trauma are associated with basal cortisol levels [32,33], and the effects may only arise during stress response. Due to the design of the SHIP study, we were not able to investigate cortisol levels after stress induction. Furthermore, single-point cortisol measurements were taken at different times of the day, and although the time of blood sampling was included, we could not account for the time since awakening, which influences the diurnal cortisol rhythm [26].

Methylation was measured in whole blood rather than in neuronal cells. However, the latter is limited to post-mortem studies and thus is not feasible for large-scale studies and application as a biomarker [34].

CM and depression were assessed as a self-report, which, especially for the retrospective CTQ, could create a bias [35]. While the cross-sectional study design only allows us to observe associations of psychopathology and HPA axis function, future longitudinal studies might allow us to draw causal conclusions.

### 3.6. Conclusion and Future Prospect

In sum, we analysed the associations between *FKBP5* methylation, *FKBP5* mRNA levels, CM, and current depressive symptoms using data from target bisulfite sequencing of the whole *FKBP5* locus. In particular, in contrast to the common approach of analysing array-based single CpG sites or pre-defined methylation regions, we analysed methylation data of the whole *FKBP5* gene. Although our sample size was not sufficient to detect small effects in a comprehensive analysis, our nominally significant results could serve as a basis for future investigations and add to the understanding of the role of *FKBP5*. Studies on larger samples are warranted to further investigate the potential mediating role of *FKBP5* methylation in the association between CM and depressive symptoms in adulthood. To facilitate future meta-analyses, all obtained results from this study are reported.

Due to its regulating role in the HPA stress response, *FKBP5* could serve as a biomarker, or as a target to develop therapy for stress-related disorders such as depression. Previous studies have reported that alterations in methylation patterns are reversible via psychotherapy [36,37], and epigenetic drug treatment for neuropsychiatric disorders is being discussed [38]. Furthermore, methylation markers may predict the outcome of therapy [39,40]. Thus, a profound understanding of the epigenetic mechanisms underlying depression could aid the development of personalised treatment and prevention.

## 4. Materials and Methods

### 4.1. Study Population

The data were obtained from the Study of Health in Pomerania (SHIP) [41,42]. SHIP is a general population project with three independently sampled cohorts and multiple follow-up assessments. For SHIP-TREND, the second SHIP cohort, 4420 adult German residents randomly selected from local registries in Western Pomerania, Germany, were assessed at the University Medicine Greifswald between 2008 and 2012. Of those 4420 subjects, a subset of 203 participants from the baseline assessment (SHIP-TREND-0), for which sequencing data were assessed, was included in the current study, hereafter referred to as SHIP-TREND.

All three SHIP cohorts were conducted in accordance with the Declaration of Helsinki, including written informed consent from all participants. The SHIP studies were approved by the Ethics Committee of the University Medicine Greifswald, Germany (approval number BB 39/08).

### 4.2. Interview and Physical Examination

Sociodemographic variables including age, sex, and smoking status were acquired by computer-assisted face-to-face interviews [41]. Smoking status was defined as never, ex- or current smoking. Physical examinations including measurements of body height and weight were performed following standard operating procedures [42] as reported in more detail elsewhere [43].

### 4.3. Assessment of Blood Measurements

Whole-blood samples were taken from the cubital vein. Blood cell counts of leukocytes, haematocrit, thrombocytes, and leukocyte subtypes were analysed using the Sysmex XT 2000 Haematology Autoanalyser (Sysmex, Kobe, Japan) according to the manufacturer’s recommendation [44]. Serum cortisol levels were measured with the Advia Centaur Cortisol Assay (Siemens Healthcare Diagnostics, Eschborn, Germany) without within-sample replication [43].

### 4.4. FKBP5 Methylation

Methylation levels of 111 CpG sites around the *FKBP5* locus, chr6:35490554-35704310 (hg19), were assessed using target bisulfite sequencing. Detailed procedures were described elsewhere [45]. Briefly, the EZ DNA-methylation Kit (Zymo Research, Irvine, CA, USA) was used for bisulfite treatment and the Illumina MiSeq (Illumina, San Diego, CA, USA) technology for sequencing. *β*-values were transformed into *M*-values, as recommended by Du et al., with β-values of zero being set to 0.5 × min (β | β > 0) to avoid *M*-values of -∞ [46].

Sites overlapping with C/T-coded SNPs with minor allele frequency (MAF) ≥ 0.05 according to the web-based annotation tool SNP Nexus [47] were excluded (n = 1 with MAF = 0.47 in this study), since they cannot be distinguished from the substitutions caused by the conversion [48]. Two CpGs with missing rates ≥ 0.05 were excluded. Three CpGs that displayed atypical *M*-value distributions were discarded (see Appendix A). After these quality control steps, 105 CpG sites remained. Finally, we excluded 71 non-variable sites, i.e., sites where the methylation beta values of all individuals were <20% or >80%, since we expected them to be less informative, resulting in 34 sites being included in the present analyses. The workflow of CpG selection is visualised in Appendix A.

### 4.5. FKBP5 mRNA Levels

*FKBP5* transcription assessment has been described elsewhere [19,49]. Briefly, mRNA was prepared with the PAXgeneTM Blood miRNA Kit (QIAGEN, Hilden, Germany). Samples with RIN < 7 were excluded [19]. RNA was reversely transcribed into complementary RNA with the Illumina TotalPrep-96 RNA Amp Kit (Ambion, Darmstadt, Germany) and labelled with biotinUTP. Transcription levels were determined with the Illumina HumanHT-12 v3 Expression BeadChip arrays (Illumina, San Diego, CA, USA). Transcription levels were quantile-normalised to standardise the intensity values between the arrays and *log2*-transformed to reduce heteroscedasticity [19,49].

### 4.6. Psychometric Assessment

Depressive symptoms were assessed using the German version of the nine-item depression module of the Patient Health Questionnaire (PHQ-9) [50]. On a four-point scale, the items assessed the nine symptoms of the A-criterion of an MDD according to the Diagnostic and Statistical Manual of Mental Disorders (DSM-IV) [51]. A summary score was calculated (range: 0–27) with higher scores indicating more severe depressive symptoms. CM was assessed with the Childhood Trauma Questionnaire (CTQ) [52] with a total of 25 items for emotional, physical, and sexual abuse as well as emotional and physical neglect. The five-point scales were added up to a summary score (range: 25–125), with higher values indicating more severe CM.

### 4.7. Genotyping

The samples were genotyped using the Illumina Human Omni 2.5 array (Illumina, San Diego, CA, USA) following the manufacturer’s recommendations and imputed using the Haplotype Reference Consortium (v1.1, build 37) reference panel. rs1360780 was imputed with imputation quality = 99.7% with a MAF = 31% in the whole genotyped sample. For more information, see Appendix A.

### 4.8. Main Statistical Analyses

Statistical analyses were performed using R version 4.1.1 [53]. For each of the four main hypotheses, two-sided linear mixed models were used to detect both positive and negative effects. The models were calculated for each CpG site separately. Next to the 34 CpG sites included in this study, we also present—for informative reasons only—the results of the 71 non-variable CpG sites in the Appendix A.

All numerical explanatory variables except for the *M*-values were normalised to a mean of 0 and a standard deviation of 1. All models included age [years], age^2^, sex [female/male], body mass index [kg/m^2^], smoking [non-/ex-/current-smoker], blood counts (thrombocytes, leukocytes) [Gpt/l], haematocrit and leukocyte subtypes (neutrophils, eosinophils, basophils, lymphocytes, and monocytes) [%] as fixed-effects covariates, and the sequencing plate identifier as a random-effect covariate. Additional covariates were added for each model:

1. Gene expression effects were tested with *FKBP5* mRNA levels as the response variable of the regression models and the methylation levels as an explanatory variable. The additional confounders, RIN and sample storage time [days], were added as well as the time of blood donation [h] using restricted cubic splines (knots at the 25th, 50th, and 75th percentile).

2. Cortisol levels were set as the response variable and the methylation levels as an explanatory variable. The covariate list was extended by the time of blood donation [h] and the fasting time [h] using restricted cubic splines (knots at the 25th, 50th, and 75th percentile) for both.

3 and 4. To assess the effects of psychometric parameters, methylation was used as the response and CTQ or PHQ-9 scores as an explanatory variable.

For each model, we visually checked that the residuals are near normally distributed. Power estimations performed with G*Power [54] were included in Appendix A.

### 4.9. Sensitivity Analysis for Genotype Interaction

To assess the effect of the SNP rs1360780, we ran sensitivity analyses for the above models 1–4, by including the interaction term rs1360780 × exposure into the model. rs1360780 was modelled as T-allele carrier vs. non-carrier [CT + TT vs. CC].

## Figures and Tables

**Figure 1 ijms-25-01485-f001:**
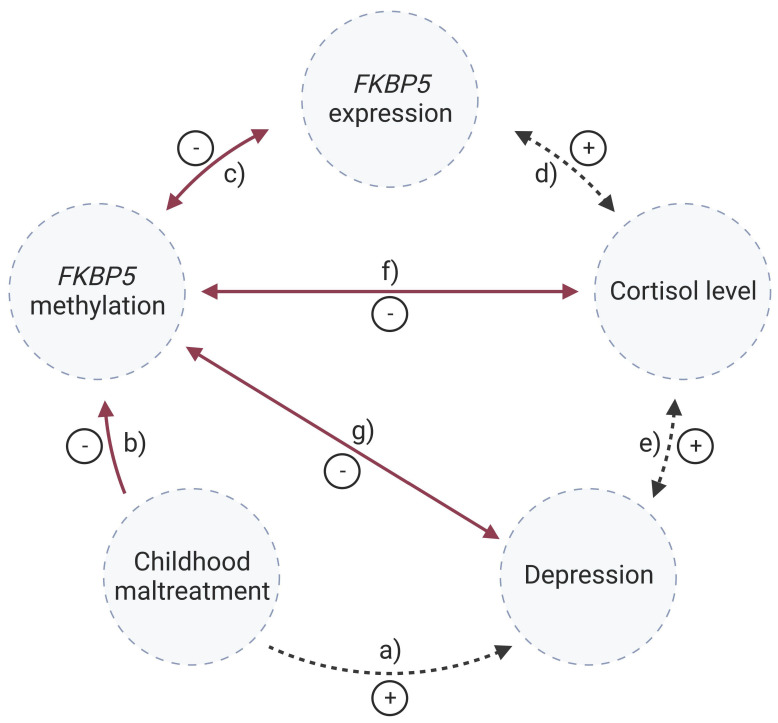
*FKBP5* methylation and psychopathology; solid arrows: hypotheses investigated in this study; dashed arrows: literature findings; +: positive associations; -: inverse associations. According to previous publications, childhood maltreatment is associated with a higher risk of adult depression [4] (**a**), and it reduces *FKBP5* methylation presumably due to high cortisol levels during childhood [10] (**b**); *FKBP5* methylation regulates *FKBP5* expression [2] (**c**). In turn, chronically increased *FKBP5* expression leads to glucocorticoid receptor resistance and subsequently to elevated cortisol levels [10,19] (**d**). High cortisol levels are associated with an increased risk for depression [20] (**e**) and demethylation of *FKBP5* via base excision repair [15] (**f**). An inverse association between *FKBP5* methylation and depression is expected (**g**). Created with BioRender.com.

**Figure 2 ijms-25-01485-f002:**
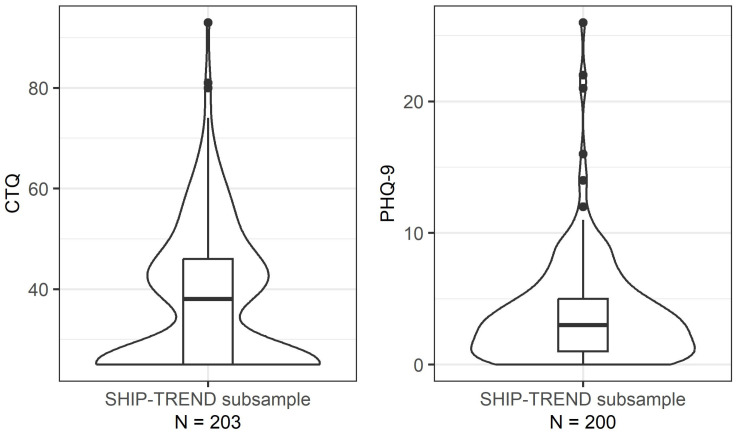
Violin and boxplot of psychological scores in the herein investigated subsample of the Study of Health in Pomerania (SHIP)-TREND; Childhood Trauma Questionnaire (CTQ) and depression module of the Patient Health Questionnaire (PHQ-9).

**Figure 3 ijms-25-01485-f003:**
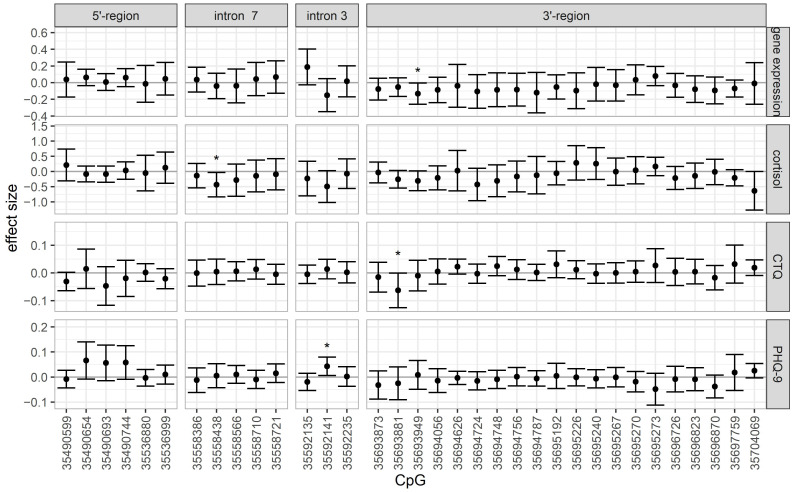
Effect sizes and 95% confidence intervals of the four regression models; * marks nominally significant findings.

**Figure 4 ijms-25-01485-f004:**
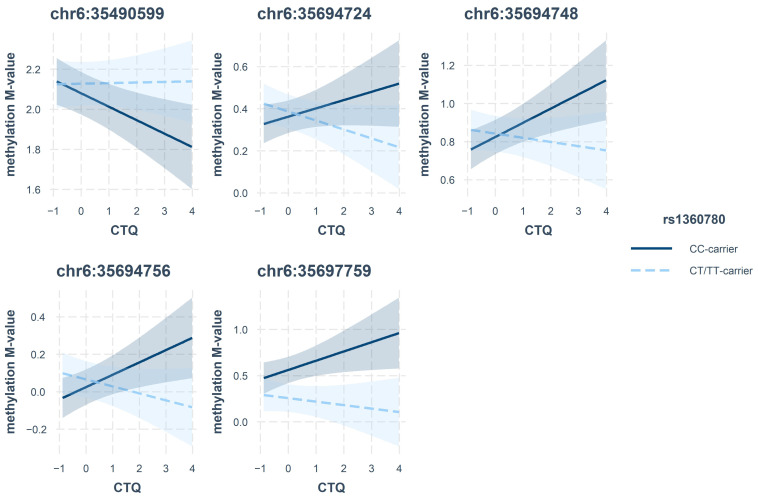
Interaction effects of rs1360780 and CTQ on methylation *M*-values (normalised).

**Figure 5 ijms-25-01485-f005:**
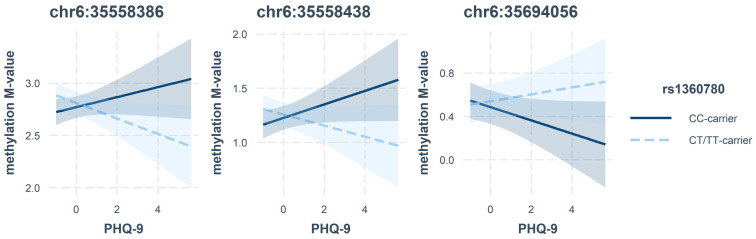
Interaction effects of rs1360780 and PHQ-9 on methylation *M*-values (normalised).

**Figure 6 ijms-25-01485-f006:**
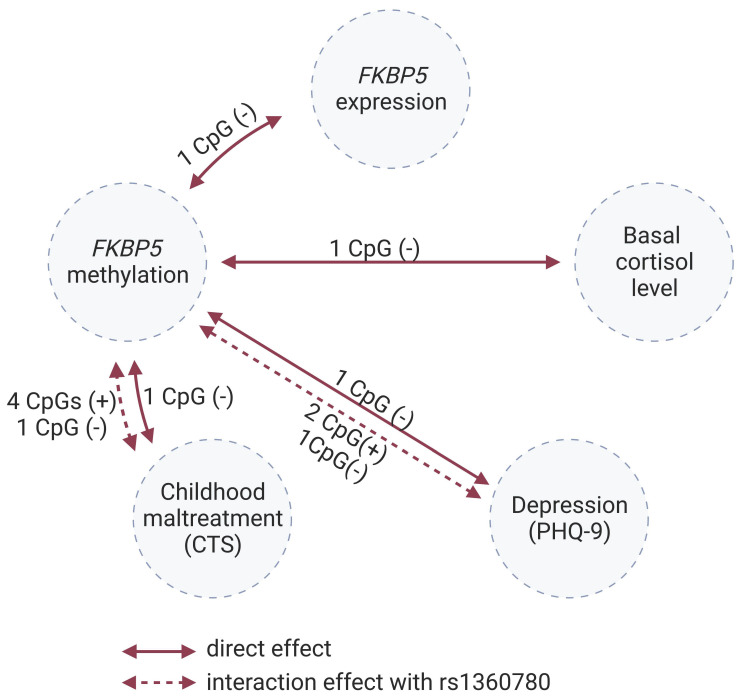
Observed associations of *FKBP5* methylation with *FKBP5* gene expression, basal cortisol levels, PHQ-9 score, and CTS score; solid arrows represent direct effects, and dashed arrows represent interaction effects with the SNP rs1360780, labelled with the number of nominally significant CpG sites for each model and the corresponding effect direction (positive (+) or negative (−)). Created with BioRender.com.

**Table 1 ijms-25-01485-t001:** Characteristics of the whole SHIP-TREND cohort and the herein analysed subsample as counts or mean ± standard deviation; NA: missing values; PHQ-9: summary score of the Patient Health Questionnaire—depression module; CTQ: summary score of the Childhood Trauma Screener.

Parameter	SHIP-TREND	Herein Analysed Subsample
*N*	4420	203
Sex	(0 × NA)	(0 × NA)
Female	2275 (51%)	97 (48%)
Male	2145 (49%)	106 (52%)
Age [years]	51.96 ± 15.46 (0 × NA)	49.00 ± 12.66
Body mass index [kg/m^2^]	28.12 ± 5.23 (7 × NA)	27.15 ± 4.53
Smoking	(22 × NA)	(1 × NA)
Never	1605 (36%)	79 (39%)
Ex	1610 (37%)	70 (34%)
Current	1183 (27%)	53 (26%)
Haematocrit	0.42 ± 0.033 (15 × NA)	0.42 ± 0.03
Thrombocytes [gpt/l]	228.10 ± 54.57 (15 × NA)	5.62 ± 1.27
Leukocytes [gpt/l]	6.18 ± 2.55 (15 × NA)	225 ± 49.6
Neutrophils [%]	58.34 ± 8.62 (31 × NA)	57.26 ± 7.44
Eosinophils [%]	2.70 ± 1.83 (201 × NA)	2.53 ± 1.56
Basophils [%]	0.49 ± 0.31 (203 × NA)	12.23 ± 0.30
Lymphocytes [%]	29.28 ± 7.57 (31 × NA)	30.60 ± 6.57
Monocytes [%]	8.78 ± 2.51 (31 × NA)	9.01 ± 2.11
mRNA	9.03 ± 0.46 (3429 × NA)	8.98 ± 0.42
Cortisol [nmol/mL]	334.4 ± 131.1 (80 × NA)	333.0 ± 135.0
PHQ-9	3.91 ± 3.59 (249 × NA)	3.87 ± 3.93 (3 × NA)
CTQ	33.34 ± 9.68 (324 × NA)	37.26 ± 13.99 (0 × NA)
rs1360780 [CC/CT + TT]	(300 × NA)	(0 × NA)
CC	2068 (50%)	103 (51%)
CT + TT	2052 (50%)	100 (49%)

**Table 2 ijms-25-01485-t002:** Nominally significant results of the main models in the subset of 34 CpGs; genetic position (chr. 6, hg19), effect size, standard error, *t*-value, degrees of freedom, *p*-value, and location of the CpG site in the gene.

Model	Position	Effect Size	S.E.	*t*-Value	d.f.	*p*-Value	Location
Gene expression	chr6:35693949	−0.133	0.065	−2.054	174	0.041	3′-end
Cortisol	chr6:35558438	−0.436	0.204	−2.137	178	0.034	intron 7
CTQ	chr6:35693881	−0.063	0.032	−1.986	180	0.049	3′-end
PHQ-9	chr6:35592141	0.043	0.019	2.303	177	0.022	intron 3

**Table 3 ijms-25-01485-t003:** Nominal significant results of the interaction models with the single-nucleotide polymorphism (SNP) rs1360780; genetic position of the CpG site (chr6, hg19), regression coefficient of the interaction term, standard error, *t*-value, degrees of freedom, *p*-value, and location of the CpG site in the gene.

Model	Position	Interaction Effect	S.E.	*t*-Value	d.f.	*p*-Value	Location
CTQ × SNP	chr6:35490599	−0.070	0.033	−2.133	179	0.034	5′-end
	chr6:35694724	0.082	0.034	2.387	181	0.018	3′-end
	chr6:35694748	0.097	0.034	2.829	179	0.005	3′-end
	chr6:35694756	0.103	0.035	2.955	179	0.004	3′-end
	chr6:35697759	0.138	0.064	2.151	181	0.033	3′-end
PHQ-9 × SNP	chr6:35558386	0.122	0.046	2.62	178	0.010	intron 7
	chr6:35558438	0.114	0.045	2.502	177	0.013	intron 7
	chr6:35694056	−0.093	0.045	−2.047	176	0.042	3′-end

## Data Availability

The data sets analysed during the current study are legally owned by the University Medicine Greifswald, represented by the steering committee of the Research Network Community Medicine. Due to data protection reasons, the data are not publicly available as the comprehensive information and high sampling fraction within the regional population could enable the identification of probands [55]. Data can be applied for upon reasonable request at https://fvcm.med.uni-greifswald.de/ (accessed on 19 December 2023). The SHIP-TREND whole-blood transcriptome data set is available at the GEO (Gene Expression Omnibus) public repository under GSE36382.

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
