# Peer review of "Methylation Patterns of the FKBP5 Gene in Association with Childhood Maltreatment and Depressive Disorders"

_ijms, 2024, doi:10.3390/ijms25031485_

Round 1
Reviewer 1 Report
Comments and Suggestions for Authors
This study has investigated the relationship of FKBP5 mRNA levels in the blood, serum levels of cortisol, childhood maltreatment, and depressive symptoms with the whole-blood methylation status. The latter has been by aid of target bisulfite sequencing of 105 CpGs at the FKBP5 locus using data from the general population-based Study of Health in Pomerania (SHIP) (N = 203). FKBP5 is a co-chaperon of the glucocorticoid receptor complex, which lowers the GR affinity to cortisol binding and delays nuclear translocation. While FKBP5 impedes the GR activity, the translocated GR-complex serves as a strong enhancer for FKBP5 transcription. Thus, FKBP5 is part of an ultra-short negative feedback loop for cortisol secretion within the hypothalamic pituitary adrenal (HPA) axis, which is the major regulatory system of the endocrine stress response. Here the investigators have examined the alterations of cortisol and FKBP5 since early stress exposure was suggested to substantially impact psychological and biological stress response mechanisms.
In total, 105 CpG sites of the FKBP5 locus (chromosome 6, hg19) were measured in this study. They hypothesized that FKBP5 methylation mediates the link between CM and adult depression by altering FKBP5 expression and thus the HPA axis stress response and that this relationship is influenced by rs1360780. They investigated 34 CpGs along the FKBP5 gene in association with CM, FKBP5 mRNA levels, cortisol, and current depressive symptoms in 203 participants of the general population-based SHIP-TREND baseline cohort. In contrast to analyzing array-based single CpG sites or pre-defined methylation regions, here methylation data of the whole FKBP5 gene were analyzed. The sample size was not sufficient to detect small effects but the authors have decided to provide data for forming the basis for future investigations and add to the understanding of the role of FKBP5.
The authors are requested to add the limitation of this study beyond the sample size.
Please add the clinical implications of these findings. What can be done even if a significant level would have been achieved?
Is there any way to prevent the methylation or reverse that, for example by behavioral therapy or the application of drugs? Please speculate.
Reviewer 2 Report
Comments and Suggestions for Authors
Abstract:
The abstract necessitates a clear statement of the research's purpose. The term "relationships" should be replaced with "correlations," as the primary objective was to identify correlations. Furthermore, please refrain from using abbreviations, such as transcribing "FKBP5." Ensure that the results are presented clearly and concisely. For instance, instead of the phrase, "As whole-gene methylation data of the FKBP5 gene has not been widely investigated, we report all obtained results to enable inclusion in future meta-analyses and to point out promising CpGs for future research beyond the ones typically analyzed," articulate the results without ambiguity and emphasize their relevance for future research. This statement could be more suitably placed towards the conclusion of the article.
Materials and Methods:
In this section, it is imperative to specify the precise location and timeframe of the research, as well as when permissions were obtained. Additionally, a flowchart illustrating the inclusion and exclusion criteria is required to enhance the clarity of the methodology.
Figure 1:
It is advisable to include the same figure in the conclusion section or modify it to reflect the actual findings and established conclusions, rather than hypotheses.
It is perplexing that the authors acknowledged on line 237 that "we could not investigate alterations in stress response" due to the use of basal cortisol levels, without prior consideration during the planning and hypothesis formulation stages. It raises questions about the judicious allocation of resources and the appropriateness of the methodology. A more suitable methodology should have been selected.
The entire manuscript, titled "Methylation Patterns of the FKBP5 Gene in Association with Childhood Maltreatment and Depressive Disorders," should be meticulously reviewed to ensure compliance with the rules of academic writing, particularly with regard to the consistent use of tenses throughout the document.
Comments on the Quality of English LanguageThe entire manuscript, titled "Methylation Patterns of the FKBP5 Gene in Association with Childhood Maltreatment and Depressive Disorders," should be meticulously reviewed to ensure compliance with the rules of academic writing, particularly with regard to the consistent use of tenses throughout the document.
Reviewer 3 Report
Comments and Suggestions for Authors
In the current study, the authors examined the relationship between FKBP5 methylation, FKBP5 mRNA levels, childhood maltreatment (CM), and current depressive symptoms using data from target bisulfite sequencing of the whole FKBP5 region. Also, associations between depression and serum cortisol levels were found.
The results are interesting. The figures reflect the results of the study. However, there are some concerns that need to be addressed.
- Which tests were used to test a data set for normality?
- Regarding the cortisol assay, did all samples measure in duplicate or triplicate in one assay? What was the variation between samples in the cortisol assay?
- Due to the female hormones (estrogen and progesterone), males and females should be separated as distinct groups.
